# Cervical Cancer Screening Participation among Women of Russian, Somali, and Kurdish Origin Compared with the General Finnish Population: A Register-Based Study

**DOI:** 10.3390/ijerph17217899

**Published:** 2020-10-28

**Authors:** Esther E. Idehen, Anni Virtanen, Eero Lilja, Tomi-Pekka Tuomainen, Tellervo Korhonen, Päivikki Koponen

**Affiliations:** 1Institute of Public Health and Clinical Nutrition, Faculty of Health Sciences, School of Medicine, University of Eastern Finland, Yliopistonranta 1, P.O. Box 1627, 70211 Kuopio, Finland; tomi-pekka.tuomainen@uef.fi; 2Finnish Cancer Registry, Unioninkatu 22, 00130 Helsinki, Finland; anni.virtanen@cancer.fi; 3Department of Pathology, University of Helsinki and HUS Diagnostic Center, Helsinki, University Hospital, Haartmaninkatu 3, 00029 HUS Helsinki, Finland; 4Department of Welfare, Finnish Institute for Health and Welfare (THL), P.O. Box 30, FI-00271 Helsinki, Finland; eero.lilja@thl.fi; 5Institute for Molecular Medicine Finland (FIMM) University of Helsinki, P.O. Box 20, 00014 Helsinki, Finland; tellervo.korhonen@helsinki.fi; 6Department of Public Health Solutions, Finnish Institute for Health and Welfare (THL), P.O. Box 30, FI-00271 Helsinki, Finland; Paivikki.koponen@thl.fi

**Keywords:** cervical cancer screening, cancer prevention, disparities, healthcare service utilization, health inequities, migrant health, women health, public health, population-based study, quantitative research

## Abstract

Migrant-origin women are less prone to cervical screening uptake compared with host populations. This study examined cervical cancer screening participation and factors associated with it in the Finnish mass screening program during 2008–2012 in women of Russian, Somali and Kurdish origin compared with the general Finnish population (Finns) in Finland. The study population consists of samples from the Finnish Migrant Health and Well-being Study 2010–2012 and Health 2011 Survey; aged 30–64 (*n* = 2579). Data from the Finnish screening register linked with other population-based registry data were utilized. For statistical analysis we employed logistic regression. Age-adjusted screening participation rates were Russians 63% (95% CI: 59.9–66.6), Somalis 19% (16.4–21.6), Kurds 69% (66.6–71.1), and Finns 67% (63.3–69.8). In the multiple-adjusted model with Finns as the reference; odds ratios for screening were among Russians 0.92 (0.74–1.16), Somalis 0.16 (0.11–0.22), and Kurds 1.37 (1.02–1.83). Among all women, the substantial factor for increased screening likelihood was hospital care related to pregnancy/birth 1.73 (1.27–2.35), gynecological 2.47 (1.65–3.68), or other reasons 1.53 (1.12–2.08). Screening participation was lower among students and retirees. In conclusion, screening among the migrant-origin women varies, being significantly lowest among Somalis compared with Finns. Efforts using culturally tailored/population-specific approaches may be beneficial in increasing screening participation among women of migrant-origin.

## 1. Introduction

Due to screening programs and through regular cervical cytological Papanicolaou (Pap) testing offered to all at-risk women as an effective preventive measure against the onset of cervical cancer [1,2,3,4]; incidence and mortality rates from cervical cancer in some higher-income countries have been significantly declining [5,6]. However, the disease’s risk is significant in women not adhering to screening recommendations [4,7,8,9], explicitly found among some migrant-origin populations [4,10,11,12,13]. Therefore, a well-organized screening program and active screening participation within the target population are imperative for comprehensive coverage and reducing the disease incidence and mortality rates [1,2,4,7].

Migrant populations have increased globally, including in Finland [14,15], and the health status of these population groups has become a public health concern, highlighting the need to strengthen healthcare systems [16,17,18,19]. Studies have revealed the underutilization of cervical cancer screening services among some migrant-origin groups compared with host populations [10,11,12,13,20,21,22,23,24,25,26]. Consequently, persisting health inequities will increase national healthcare expenditure [27]. Studies have linked some barriers to cervical cancer screening participation at the individuals and the screening system levels [28,29,30]. 

Some individual barriers to cervical cancer screening participation include low socioeconomic status, being unmarried, and unemployed, illiteracy, and limited language skills [11,21,22,25,30,31]; inadequate understanding or unawareness about screening/purpose, cancer risk in the host country/country of origin, and fear of cancer or screening [30,31,32,33,34,35]; cultural/religious beliefs; and unpleasant screening experiences, such as pain, and embarrassment owing to Female Genital Multilation (FGM), and/or obesity [26,36,37,38,39]. Some other individual barriers are being an older/nulliparous woman, not using healthcare services related to gynecological, reproductive, or pregnancy [21,30,40,41,42], and issues related to migration and residential areas [22,25,26,33,43]. System barriers include women’s distrust in healthcare authorities and personnel, unavailability of female screeners, inaccessibility to healthcare, and interpretation services [25,32,33,35,39,43].

In Finland, municipalities are obliged to offer free-of-charge cervical cancer screening to eligible women aged 30–60, with a five-year interval [44]. Additionally, some municipalities invite women aged from 25 to 65. Eligible women are identified from the Finnish National Population Registry covering all Finnish citizens and foreign citizens residing in Finland on a permanent or temporary basis (excluding only undocumented migrants), and personal invitations are sent by mail [44]. The invitation coverage is almost 100% among women aged 30–60 [44]. Presently, the overall participation rate in the cervical screening program is about 70% [44], although the aim is to achieve 80–85% screening coverage in the population [44]. In addition to the organized program, opportunistic screening, i.e., the testing of non-symptomatic persons outside the program, is widespread in the country [45]. Opportunistic screening is not included in any national register [46].

Previous studies in Finland demonstrated disparities and lower cervical cancer screening participation among migrant-origin women [20,21] and those with a non-native mother tongue [30,46,47,48], compared with the general Finnish population (hereafter referred to as Finns). However, register-based studies about cervical cancer screening participation in the mass screening program among various migrant-origin populations are limited. This study examined cervical cancer screening participation and factors associated with it in the Finnish mass screening program during 2008–2012 in women of Russian, Somali, and Kurdish origin compared with the general Finnish population in Finland. 

## 2. Material and Methods 

### 2.1. Study Population

This study population includes women from the samples of the Migrant Health and Well-being Study 2010–2012 (Maamu) [49] and the Health 2011 Survey [50], both of which were carried out by the Finnish Institute for Health and Welfare (THL). The sample from the Maamu study comprises migrant-origin populations of 1998 Russians (1230 women), 1963 Somalis (1020 women), and 1948 Kurdish-origin (819 women) (briefly, hereafter referred to as Kurds). The study stratified random sample was drawn from the Finnish National Population Registry; inclusion criteria were as follows: age 18–64, residence in one of the six cities (Helsinki, Vantaa, Espoo, Tampere, Turku, and Vaasa) with a high proportion of the migrant-origin population, and residence for at least a year in Finland. The country of birth was Somalia, Russia or the former Soviet Union, and Iran or Iraq. The native language was Russian or Finnish for the Russian group and Kurdish Sorani for the Kurdish-origin group.

The reference population (Finns) consists of women from the Health 2011 Survey sample, including the same age groups and cities as in the Maamu Study. We identified a small portion of women from the study groups who had no information about the invitation to the mass screening; we excluded them from further analysis. This study sample was narrowed to women aged 30–64 and those migrants who arrived in Finland before 2008. Thus, all women included in the analyses were: 816 Russians, 523 Somalis, 451 Kurds, and 789 Finns, who had received at least one invitation from the organized screening program (Figure 1).

### 2.2. Data Sources and Variables

We employed data from the: (1) National Population Register [51], (2) Mass Screening Registry [52], (3) Care Register for Health Care [53], (4) Medical Birth Register [54], (5) Register of Induced Abortions [55], (6) Statistics Finland [56] and, (7) Social insurance institution of Finland (Kela), [57]. The data from different sources for persons in the Maamu and Health 2011 Survey samples were linked using the unique personal identity codes [58] given to Finland’s legal residents.

### 2.3. Study Outcomes

The study outcome measure was participation in the organized cervical cancer screening program among all eligible women invited during 2008–2012 in Finland. In Finland, the invitation to screening is usually valid for a year, after these women are regarded as non-attendees.

### 2.4. Variable Definitions

Invitation and participation data for the screening program came from the Mass Screening Registry [52]; they were dichotomously coded as yes vs. no; yes indicated a positive response to the latest invitation 2008–2012. We obtained data on age from the population register; our three categories were 30–39, 40–49, and 50–64. Education data came from Statistics Finland and was dichotomously coded as at least high school or equivalent vs. upper secondary or less by 2012.

Marital status data was from the population register on the date the sample was drawn in 2010; the status was dichotomously coded as married or in a civil union vs. any other or unknown. The year of migration to Finland came from the population register being dichotomously coded as 1970–1997 vs. 1998–2007. Employment status data were from Statistics Finland for the invitation year or the closest available year (2009, 2010, or 2011). When unknown, the data were supplemented with information on Kela benefits from 2010–2012. This data was coded into five categories: employed, childcare at home, student, retired, and unemployed/unknown.

Information on the item having moved from one municipality to another in Finland came from the Population Register and was dichotomously coded for 2008–2011 as a change in residence municipality vs. no change during that period. This information was only available for women with a migrant background, who are likely to move more often than the general population. Information on the item having stayed abroad for over one year from Finland was obtained from the Population register and dichotomously coded as having stayed abroad for at least one year vs. less or not at all, between 2009 and 2011.

The number of births (the number of deliveries an individual had) in Finland came from the Medical Birth Register for 1987–2012 and was coded as none, 1 or 2, and 3 or more births. Only those births before the screening invitation year were considered. The number of abortions in Finland was obtained from the Register of Induced Abortions and was dichotomously coded as follows: none vs. one or more, covering 1983–2012. Only abortions before the screening invitation year were considered.

Hospital care in Finland was identified from the Care Register for Health Care as having been in secondary healthcare as inpatient or outpatient: these recorded visits were categorized as: no care; pregnancy or birth-related care (International Classification of Diseases (ICD 10) codes O00–O08, O10–O16, O20–O29, O30–O48, O60–O75, O80–O92, O94–O99); other gynecological care (ICD10: N70–77, N80–N98, and gynecological cancers C51–C58, D06, D07); and hospital care for any other reason. Only hospital care from 1994 until the year before the screening invitation was considered.

### 2.5. Statistical Analyses 

For statistical analyses, we utilized SAS 9.3 and SUDAAN 11.0.3 software [59]. The study population’s age-adjusted main descriptive characteristics were explored by country of origin. Furthermore, logistic regression was used, adjusting for age to explore factors associated with participation. The study results are shown as odds ratios (ORs), 95% confidence intervals (CIs), *p*-values, and model-adjusted screening participation percentages. We used the Akaike information criterion (AIC) [60] to determine the best predictive model for screening participation. In the model selection, we considered all the main effects and how these interacted with the country of origin.

For the final model, the selected variables were country of origin, age, education, employment status, staying abroad, hospital care in Finland, and the interaction between the country of origin, and year of migration. The statistical significance was assessed with the Satterthwaite-adjusted F-value. Age-adjusted proportions were calculated using predictive margins [61]. In all the analyses, we applied Finite Population Correction (FPC) to the migrant-origin groups and inverse probability weights (IPWs) [62], due to unequal sampling probabilities within the study groups. 

### 2.6. Ethical Considerations 

The coordinating ethics committee of the Hospital District of Helsinki and Uusimaa in Finland approved the use of register data for both the Maamu Study and Health 2011 Survey samples, including the linkage of data from different registers (decisions #325/13/03/00/2009 and 45/13/03/00/11). All data was analyzed and stored at THL, following THL data safety regulations and complying with the European Union General Data Protection Regulation [63].

## 3. Results 

### 3.1. Characteristics 

Table 1 displays an overview of the characteristics of the study populations. Women of Russian origin had a higher proportion of high education level (79%) and employment (60%), compared with Kurds (33% and 32%), Somalis (19% and 26%); the highest percentages were those of the Finns (87% and 81%). The Kurds had the highest proportion of being married or in a civil union (76%), compared to Russians (54%), Somalis (65%), and Finns (48%).

### 3.2. Screening Participation in the Age-Adjusted Analysis

The age-adjusted screening participation rate among women invited to the screening was highest amongst the Kurds (69%), followed by Finns (67%) and Russians (63%); it was markedly lowest among Somalis (19%).

Among the Russians, a higher education level (OR = 1.52; 95% confidence interval (CI): 1.06–2.17), and having had hospital care related to pregnancy/birth (OR = 1.97; 95% CI: 1.29–3.02), or for other gynecological reasons (OR = 2.09; 95% CI: 1.13–3.86) compared to no history of hospital care in Finland were associated with screening participation. Additionally, increased participation with advanced age was observed. Retirees participated less (OR = 0.39; 95% CI: 0.17–0.87) compared with employed women. 

Among the Somalis, higher screening participation was associated with a recent migration to Finland (OR = 1.56; 95% CI: 1.10–2.21), and those aged 40–49 (OR = 1.49; 95% CI: 1.03–2.16); age 30–39 as the reference. Participation was lower among students (OR = 0.37; 95% CI: 0.22–0.62), with employed women as the reference. Participation decreased as the number of births increased (OR = 0.57; 95% CI: 0.36–0.92).

Among the Kurds, factors associated with higher screening participation were age 40–49 (OR = 1.64; 95% CI: 1.30–2.06) as well as hospital care relating to pregnancy/birth (OR = 3.54; 95% CI: 1.92–6.53), gynecological (OR = 3.01; 95% CI: 1.49–6.08), or for other reasons (OR = 2.70; 95% CI: 1.42–5.13), compared to no history of hospital care in Finland. Other factors were having lived in one municipality in Finland compared to having moved from one municipality to another (OR = 1.79; 95% CI: 1.16–2.75); being married or in a civil union (OR = 1.57; 95% CI: 1.24–1.99), and higher education level (OR = 1.41; 95% CI: 1.13–1.76). Lower participation was associated with being a student (OR = 0.68; 95% CI: 0.48–0.95), retired (OR = 0.54; 95% CI: 0.37–0.78), or unemployed/unknown (OR = 0.66; 95% CI: 0.51–0.86). 

Among the Finns, older age (50–64), (OR = 1.96; 95% CI:1.35–2.83), higher education level (OR = 1.87; 95% CI: 1.20–2.89), and hospital care due to other gynecological reasons (OR =2.77; 95% CI: 1.40–5.49), as well as staying abroad for less than a year (OR = 4.86; 95% CI: 1.62–14.57) compared to staying abroad for at least a year during the study period were associated with higher screening participation (Table 2).

### 3.3. Screening Participation in the Multiple-Adjusted Analysis

Table 3 provides the results of the multiple-adjusted analysis, with all three migrant-origin groups and Finns combined. When adjusted for age, education, employment status, staying abroad, and hospital care in Finland, screening participation was highest among the Kurds (OR = 1.37; 95% CI: 1.02–1.83) and lowest among the Somalis (OR = 0.16; 95% CI: 0.11–0.22), with Finns as the reference. Participation rates among Russians were similar to those of the Finns (OR = 0.92; 95% CI: 0.74–1.16).

Among all the groups, the most substantial factor associated with increased screening likelihood was hospital care (related to pregnancy or birth, or other gynecological reasons, and other reasons). Other factors were older age, higher education level, and having not stayed abroad for over a year. Concerning employment status, participation was lower among students and retirees compared with employed women.

## 4. Discussion

This population register-based study examined cervical cancer screening participation and factors associated with it in the Finnish mass screening program during 2008–2012 in women of Russian, Somali, and Kurdish origin compared with the general Finnish population (Finns) in Finland. Our study revealed differences in the screening uptake amongst migrant-origin women, consistent with previous studies [10,11,12,13,22,23,24,25,26,30,46,64]. Our study is the first population register-based study to report participation in the Finnish mass screening program among these specific migrant-origin groups compared with the general Finnish population in Finland to the best of our knowledge. These findings contribute to existing knowledge and enhance some understanding of cervical cancer screening participation among migrant-origin populations. Sequentially, this can guide policymakers in developing cervical cancer screening protocols to improve participation among migrant-origin populations. 

In our study, the participation rate was highest among the Kurds, Russians participated similarly to the Finns, and the rate was clearly lowest among the Somalis, even after adjusting for age, education, employment status, staying abroad, and hospital care in Finland. The low cervical cancer screening participation rate observed amongst Somalis accords with earlier results based on self-reported Pap-smear uptake in Finland [20,21]. The results among the Kurds and Russians differed from those of the previous studies. The relatively good Finnish/Swedish (official Finnish languages) skills among the Russians and Kurds compared with Somalis can partly explain these disparities, as shown in our previous self-reports of population-based study based among these groups [21], and a qualitative study among other migrant groups in Finland [30]. 

In Finland, healthcare services like screening are offered free-of-charge (except with a nominal fee in few cases) “to all legal inhabitants and their families, irrespective of their cultural background and socioeconomic conditions” [65]. Nevertheless, migrants are a heterogeneous population, and their socioeconomic status can vary depending on their occupation, educational level, and reason for migration. The differences in healthcare systems, social structures, and cultures in the migrants’ new environment compared to their previous societies may impact their utilization of healthcare services such as screening [66,67]. In our study, the migrants’ countries of origin and the purpose of their migration to Finland also differ in several ways [49]. A more extended stay in the immigrants’ host country might enable them to get acquainted with the healthcare system. The health care system in Russia may be more similar compared to the Finnish system than the health care services available in Somalia and Iran, or Iraq.

The Somalis’ markedly low screening rate is consistent with those in international qualitative studies [35,38,39]. These studies demonstrated possible barriers hindering screening participation such as childcare, cultural or religious beliefs, fear, inadequate understanding of screening, limited language skills, absence of female healthcare personnel, mistrust in healthcare authorities, unpleasant screening experiences such as pain, and embarrassment due to FGM practices [35,38,39]. Nonetheless, in our study, such variables were missing, as our main goal was to compare the migrant-origin women with the Finns, using register data for participation in the organized cervical cancer screening in Finland.

Generally, both risk and incidence of cervical cancer are significant in women not adhering to screening recommendations [4,7,8,9], explicitly found among some migrant-origin populations [4,10,11,12,13]. The incidence and mortality of cervical cancer are low in Finland [44], but current overall incidence rates might camouflage the risk of cancer in certain subgroups of the increasingly heterogenic population. The low screening participation among the Somali-origin women is a cause for concern when considering the disease’s risk in the country of origin [68]. There is a strong need for analyses of cancer incidence stratified by immigrant groups.

The increased screening participation rate observed among older women agrees with other studies [37,46,47,48,69,70]; contradictory reports also exist [11,24]. Higher screening participation with increasing age might stem from a better understanding of the importance of cancer screening among older women [37] or the fact that they have received several screening invitations and are thus more familiar with the concept. On the other hand, overall, lower screening participation among younger women has been connected with higher uptake in opportunistic testing [46]. Younger women may skip the invitation to organized screening if they have recently been tested during health care visits connected to sex counselling and contraception [71]. The higher screening participation observed among younger Kurds compared with the Finns is consistent with previous studies [46,47,48]. Younger women in the Kurdish group use less often contraceptive methods requiring medical follow-up than younger women in the general population [71]; they are likely less often offered opportunistic screening. 

The association between higher education and higher screening participation is in line with previous national findings from Finland [46,47] and self-reported screening test uptake in Finland, here mainly observed among the Kurds [21], and with other international studies [11,24,37]. Higher education and health literacy may thus be associated [72,73]. Additionally, when we used employees as a reference, no differences between the employed and women taking care of children at home emerged, except for students and retirees, who exhibited lower screening participation. In Finland, people can receive a retirement pension at around age 65 [74]. So, the reasons for retirement among the women studied include mainly severe long-term illness or disability, which can influence screening participation. The association of being married or in a civil union with higher screening participation is consistent with previous findings [11,25,47].

The higher likelihood of screening participation observed among all the groups associated with healthcare services utilization related to gynecological reason, pregnancies, or births history agrees with other studies [21,40,41,42]. Contact with healthcare might enable women to be more acquainted with the healthcare system and obtain disease prevention-related information, thus facilitating healthcare services use such as screening participation [43]. Finally, the association of absence from Finland for an extended period with decreased screening participation is apparent as the screening invitation is sent as a personal invitation letter, which is usually valid for a year [44].

### Limitations and Strengths

This study has some limitations. Firstly, it has examined cervical cancer screening participation in the Finnish screening program among only these groups and specific cities in Finland; we cannot generalize these results to all migrant-origin women populations. Secondly, the study used data based on registered information on women invited to the screening within a specific period. We are uncertain whether all the women invited received these mailed invitations; the population register may contain inaccurate addresses. Similarly, missing items in the register data may have caused some bias; unknown education or employment status is an example. Thirdly, this study focused on organized screening uptake but information on the use of opportunistic testing would have provided valuable background information as an explanatory factor to non-participation in the organized program.

This study’s key strengths are that it utilized national registries, all of which have proven data quality and coverage in Finland [75,76,77]. Self-reported screening uptake may be vulnerable to recall bias [78]; in contrast, register-based data can provide more reliable information on screening in the organized program. Furthermore, population-based random survey samples were used to identify women in the three groups, which are among the largest migrant-origin populations in Finland, and the cities selected covered a significant share of these populations. The size of the city-specific sub-samples was determined according to the size of the foreign-born population living in a particular city [49].

## 5. Conclusions

Despite the free-of-charge mass cervical cancer screening organized in Finland, our study demonstrated that screening participation among the migrant origin women varies, being significantly lowest among Somalis compared with women in the general Finnish population (Finns). Efforts using culturally tailored/population-specific approaches may be beneficial in increasing screening participation among women of migrant origin. More attention is needed to raise awareness about the importance of screening and preventive care in certain groups. These groups comprise women of migrant backgrounds, low socioeconomic status, younger women, students, retirees, and not using healthcare services related to gynecological or/and reproductive reasons. The markedly low participation rate observed among Somali-origin women is a cause for concern and requires further attention. More broadly, qualitative research is needed to explore this group’s perceptions regarding the screening and factors related to social relationships, religion, or culture, which cannot be studied with register-based data.

## Figures and Tables

**Figure 1 ijerph-17-07899-f001:**
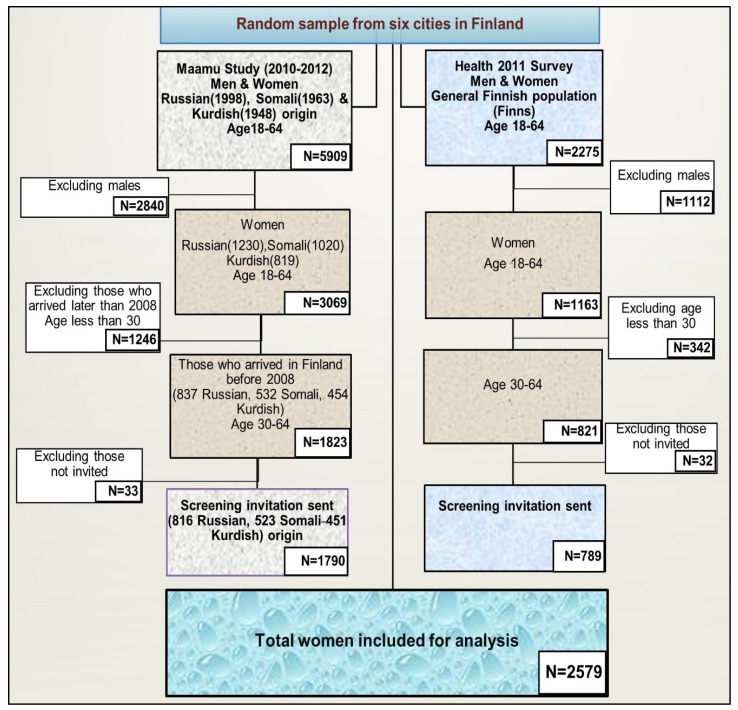
Flow chart of the study population. (Six cities: Helsinki, Espoo, Vantaa, Turku, Tampere, and Vaasa).

**Table 1 ijerph-17-07899-t001:** Mass screening participation in the study population and baseline characteristics of the women meeting the inclusion criteria (aged 30–64) and the migrant origin groups who have moved to Finland between the years 1970 and 2007) by country of origin: weighted and age-adjusted proportions.

Characteristics (*n* = 2644)		Russian (*n* = 837)		Somali (*n* = 532)		Kurdish (*n* = 454)		Finns (*n* = 821)	
Mass screening participation									
Variable/description		*%*	95%CI ^1^	%	95%CI	%	95%CI	%	95%CI
	No invitation	2.6	1.7–4.1	2.1	1.3–3.4	0.9	0.4–1.8	3.5	2.5–4.9
	Invited, not participated	34.0	30.8–37.4	79.0	76.2–81.6	30.2	28.1–32.4	29.9	26.8–33.1
	Invited, participated	63.3	59.9–66.6	18.9	16.4–21.6	68.9	66.6–71.1	66.6	63.3–69.8
Age									
	30–39	29.1	26.0–32.3	44.0	40.8–47.3	43.5	41.0–45.9	28.7	25.8–31.9
	40–49	34.3	31.0–37.6	36.2	33.1–39.4	39.9	37.5–42.4	28.5	25.5–31.7
	50–64	36.7	33.4–40.1	19.7	17.3–22.4	16.6	14.9–18.5	42.8	39.4–46.2
Education ^2^									
	High school or higher	78.7	75.7–81.4	18.8	16.4–21.4	33.0	30.7–35.3	87.0	84.7–89.1
	Less than high school/unknown	21.3	18.6–24.3	81.2	78.6–83.6	67.0	64.7–69.3	13.0	10.9–15.3
Marital status									
	Married or in a civil union	53.7	50.1–57.1	65.4	62.2–68.5	76.5	74.4–78.6	48.5	45.0–51.9
	Other/Unknown	46.3	42.9–49.9	34.6	31.5–37.8	23.5	21.4–25.6	51.5	48.1–55.0
Employment status									
	Employed	60.1	56.6–63.5	26.0	23.2–29.0	32.1	29.9–34.5	81.1	78.3–83.5
	Childcare at home	5.6	4.2–7.4	5.3	4.0–6.9	5.8	4.8–7.1	1.7	1.0–2.9
	Student	6.1	4.6–8.2	22.1	19.8–24.5	11.9	10.6–13.4	2.5	1.5–3.9
	Retired	3.7	2.6–5.2	6.6	5.0–8.5	13.2	11.5–15.2	5.1	3.9–6.7
	Unemployed/unknown	24.5	21.7–27.6	40.0	36.9–43.2	36.9	34.4–39.4	9.6	7.9–11.7
Year of migration									
	1970–1997	52.0	48.5–55.4	66.2	63.0–69.2	43.7	41.3–46.2		
	1998–2007	48.0	44.6–51.5	33.8	30.8–37.0	56.3	53.8–58.7		
Had moved from one municipality to another								Na ^3^	
	No	94.6	92.5–96.1	97.0	95.4–98.0	95.8	94.9–96.6		
	Yes	5.4	3.9–7.5	3.0	2.0–4.6	4.2	3.4–5.1		
Had stayed abroad for over one-year ^4^									
	No	97.4	96.0–98.3	92.3	90.2–94.0	93.1	91.5–94.4	97.7	96.4–98.6
	Yes	2.6	1.7–4.0	7.7	6.0–9.8	6.9	5.6–8.5	2.3	1.4–3.6
Number of births given in Finland									
	None	56.3	53.1–59.5	36.8	34.0–39.7	42.2	40–44.5	42.4	39.2–45.8
	1–2	40.8	37.6–44.0	15.5	13.4–18.0	47.3	44.9–49.6	45.3	41.9–48.8
	3 or more	2.9	2.0–4.4	47.7	44.7–50.7	10.5	9.3–11.8	12.2	10.1–14.8
Number of abortions in Finland									
	None	81.1	78.2–83.7	90.3	88.3–92.0	77.2	75.1–79.2	82.1	79.2–84.6
	1 or more	18.9	16.3–21.8	9.7	8.0–11.7	22.8	20.8–24.9	17.9	15.4–20.8
Hospital care in Finland ^5^									
	None	20.1	17.4–23.0	4.7	3.4–6.5	4.6	3.5–6.1	14.8	12.5–17.3
	Pregnancy/birth-related care	41.8	38.6–45.0	61.3	58.4–64.2	59.3	57.0–61.6	38.6	35.4–41.9
	Other gynecological reason	10.2	8.3–12.5	10.0	8.1–12.4	9.4	8.0–11.0	11.7	9.7–14.0
	Other reason	28.0	25.1–31.1	23.9	21.4–26.7	26.7	24.4–29.1	35.0	31.9–38.2

^1^ 95% CI = 95% confidence interval; ^2^ High school education registered in Finland; ^3^ Na = not available; ^4^ Had stayed abroad for a period of over one year between the years 2008–2011; ^5^ Hospital care in Finland = inpatient or outpatient care in secondary or tertiary health care for the specified reasons.

**Table 2 ijerph-17-07899-t002:** Age-adjusted participation in mass cervical cancer screening by country of origin (includes only women invited to mass screening).

Total Women (*n* = 2579)		Russian (*n* = 816)			Somali (*n* = 523)			Kurdish (*n* = 451)			Finns (*n* = 789)		
Mass screening participation Invited, participated		%	95% CI ^1^		%	95% CI		%	95% CI		%	95% CI	
		**63.3**	59.9–66.6		**18.9**	16.4–21.6		**68.9**	66.6–71.1		**66.6**	63.3–69.8	
Variable/description		%	OR ^2^ (95%CI)	*p*	%	OR (95%CI)	*p*	%	OR (95%CI)	*p*	%	OR (95%CI)	*p*
Age													
	30–39	60.1	1.00	0.120	16.8	1.00	0.027	63.9	1.00	<0.001	62.2	1.00	0.001
	40–49	67.0	1.35 (0.93–1.96)		23.1	1.49 (1.03–2.16)		74.4	1.64 (1.30–2.06)		67.5	1.26 (0.86–1.85)	
	50–64	68.4	1.44 (1.00–2.07)		14.7	0.85 (0.53–1.37)		67.5	1.17 (0.89–1.54)		76.3	1.96 (1.35–2.83)	
Education ^3^													
	Upper secondary/less	57.9	1.00	0.023	19.4	1.00	0.300	66.1	1.00	0.002	57.4	1.00	0.005
	High school or more	67.6	1.52 (1.06–2.17)		15.9	0.78 (0.50–1.24)		73.3	1.41 (1.13–1.76)		71.3	1.87 (1.20–2.89)	
Marital status													
	Other/unknown	64.6	1.00	0.612	19.3	1.00	0.736	61.0	1.00	<0.001	69.3	1.00	0.848
	Married/in a civil union	66.3	1.08 (0.80–1.46)		18.4	0.94 (0.66–1.34)		70.9	1.57 (1.24–1.99)		69.9	1.03 (0.76–1.40)	
Employment status													
	Employed	66.9	1.00	0.195	22.7	1.00	0.001	74.0	1.00	0.001	70.8	1.00	0.399
	Childcare at home	62.0	0.80 (0.42–1.54)		27.4	1.28 (0.64–2.56)		75.5	1.08 (0.7–1.68)		71.3	1.02 (0.29–3.63)	
	Student	65.7	0.95 (0.48–1.88)		9.9	0.37 (0.22–0.62)		65.9	0.68 (0.48–0.95)		51.1	0.42 (0.15–1.18)	
	Retired	44.1	0.39 (0.17–0.87)		23.6	1.05 (0.48–2.31)		60.6	0.54 (0.37–0.78)		66.2	0.81 (0.42–1.56)	
	Unemployed/unknown	66.0	0.96 (0.67–1.37)		20.7	0.89 (0.56–1.39)		65.4	0.66 (0.51–0.86)		64.3	0.74 (0.43–1.26)	
Year of migration													
	1970–1997	68.0	1.00	0.129	16.3	1.00	0.013	67.4	1.00	0.358			
	1998–2007	62.6	0.79 (0.58–1.07)		23.2	1.56 (1.10–2.21)		69.6	1.11 (0.89–1.38)				
Had moved from one municipality to another											Na ^4^		
	Yes	60.0	1.00	0.495	26.2	1.00	0.371	55.9	1.00	0.008			
	No	65.8	1.28 (0.63–2.63)		18.5	0.64 (0.24–1.72)		69.3	1.79 (1.16–2.75)				
Had stayed abroad for over one-year ^5^													
	Yes	45.2	1.00	0.076	15.8	1.00	0.560	60.9	1.00	0.097	33.4	1.00	0.005
	No	66.0	2.36 (0.91–6.09)		18.9	1.24 (0.60–2.57)		69.3	1.46 (0.93–2.27)		70.4	4.86 (1.62–14.57)	
Number of births given in Finland													
	None	63.6	1.00	0.249	24.2	1.00	0.045	68.7	1.00	0.342	70.0	1.00	0.974
	1–2 births	68.9	1.27 (0.88–1.83)		20.6	0.81 (0.46–1.42)		69.9	1.04 (0.80–1.35)		69.2	0.96 (0.68–1.36)	
	3 births or more	56.9	0.75 (0.32–1.76)		15.6	0.57 (0.36–0.92)		64.7	0.83 (0.59–1.17)		69.2	0.96 (0.57–1.6)	
Number of abortions in Finland													
	None	64.9	1.00	0.455	18.9	1.00	0.736	68.4	1.00	0.636	70.1	1.00	0.491
	1 or more	68.3	1.17 (0.78–1.75)		17.2	0.90 (0.47–1.70)		69.6	1.06 (0.84–1.33)		67.2	0.87 (0.59–1.29)	
Hospital care in Finland ^6^													
	None	55.0	1.00	0.010	14.9	1.00	0.061	41.5	1.00	<0.001	62.4	1.00	0.025
	Pregnancy/birth- related care	70.5	1.97 (1.29–3.02)		16.7	1.15 (0.50–2.66)		71.1	3.54 (1.92–6.53)		66.8	1.21 (0.73–2.02)	
	Other gynecological reason	71.7	2.09 (1.13–3.86)		27.9	2.23 (0.88–5.70)		67.7	3.01 (1.49–6.08)		82.0	2.77 (1.40–5.49)	
	Other reason	62.7	1.38 (0.89–2.14)		22.9	1.71 (0.69–4.24)		65.3	2.70 (1.42–5.13)		70.5	1.45 (0.87–2.41)	

^1^ 95% CI = 95% confidence interval; ^2^ OR = odds ratio; ^3^ high school education registered in Finland; ^4^ Na = not available; ^5^ had stayed abroad between the years 2008–201; ^6^ Hospital care in Finland = inpatient or outpatient care in secondary or tertiary health care for the specified reasons.

**Table 3 ijerph-17-07899-t003:** Multiple adjusted model ^1^ for participation in mass cervical cancer screening.

Total *n* of Women (*n* = 2579)			
Variable/Description		OR (95% CI) ^2^	*p*
Study groups			
	Finns	1.00	<0.001
	Russian	0.92 (0.74–1.16)	
	Somali	0.16 (0.11–0.22)	
	Kurdish	1.37 (1.02–1.83)	
Age			
	30–39	1.00	0.003
	40–49	1.30 (1.04–1.62)	
	50–64	1.56 (1.20–2.02)	
Education ^3^			
	Upper secondary or less	1.00	0.006
	High school or more	1.37 (1.10–1.70)	
Employment status			
	Employed	1.00	0.006
	Childcare at home	0.95 (0.61–1.48)	
	Student	0.59 (0.43–0.83)	
	Retired	0.60 (0.40–0.89)	
	Unemployed/unknown	0.82 (0.65–1.03)	
Year of migration			
	1970–1997	1.00	0.090
	1998–2007 among Russian	0.91 (0.66–1.26)	
	1998–2007 among Somali	1.69 (1.08–2.65)	
	1998–2007 among Kurdish	1.21 (0.82–1.77)	
Had stayed abroad for over one year			
	Yes	1.00	0.008
	No	1.87 (1.18–2.96)	
Hospital care in Finland ^4^			
	None	1.00	<0.001
	Pregnancy and birth-related care	1.73 (1.27–2.35)	
	Any other gynecological reason	2.47 (1.65–3.68)	
	Any other reason	1.53 (1.12–2.08)	

^1^ Including study group, age, education, living abroad, hospital care in Finland and employment status, and using the year of migration separately for each migrant group; ^2^ OR = odds ratio, 95% CI = 95% confidence interval; ^3^ high school education registered in Finland; ^4^ Hospital care in Finland = inpatient or outpatient care in secondary or tertiary health care for the specified reasons.

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
