# Peer review of "Cervical Cancer Screening Participation among Women of Russian, Somali, and Kurdish Origin Compared with the General Finnish Population: A Register-Based Study"

_ijerph, 2020, doi:10.3390/ijerph17217899_

Round 1
Reviewer 1 Report
The authors addressed an important weakness in that the study was only in 6 urban centers and it would be of interest to compare patients in rural vs urban setting. Having said that I think it is a well done paper and will hopefully lead to some modifications or programs to improve participation in the Somali population in particular.
1. Study looks at a cohort of 3 different migrant populations Russian, Somali and Kurds, compared a baseline native (Finnish) population. While access to cervical cancer screening is a covered/mandatory benefit for all within this population there are significantly large disparities among the 4 populations especially within the Somali population compared to the rest. The authors try to look at various factors that may influence positive or negative factors that could be part of the issue. While this work is not "original" I feel it gives more information and issues that we need to address within an increasing Global healthcare and disparities that continue to lead to sub adequate care within different populations.
2. I think that a strength of this study is that it looked at a rather diverse population of patient regarding previous engagement with the health care system, age, education, marital status employment, While the cohort numbers were not extremely large I feel that they do give a significant number to be relevant. The major weakness that I saw and the authors themselves addressed in their manuscript was that the study population came from 6 urban environments and did not look at a comparable rural cohort of patients which I think could have significantly added more strength to their study but I do not feel that this is reason for rejection or even additional data collection at present. I think it does lead to a further larger study in the future that would address this area of concern.
3. I think that this study can stand on its merits. There is a large amount of data analysis that is done but feel in a study of this type it is important to try and tease as much data as possible. The authors use of English is excellent from my opinion and frankly have seen poorer use of English in papers that originate in US and other English speaking countries. If I had any recommendation for improvement it would be towards usage of both a rural and urban population and see any significant difference but again do not see it as a reason to delay or reject this manuscript. I live in an area with a very large Somali migrant population and unfortunately see disparities similar to this in our population here. I would hope that papers such as this add more light to this topic and more general information as to ways that we may consider addressing some of these disparities.
Author Response
Please, see the attachment.

Reviewer 2 Report
The research methodology is well described and the results of data analyses are clearly presented in the paper. It's a very informative article.
It appears that younger Kurdish women in the study reported a higher participation rate than the Finnish women in the similar age/education groups. Some of the differences are even statistically significant. The authors are recommended to provide more discussions on whether those findings were unexpected or understandable in the context of the migrant-origin women in Finland.
Author Response
Please, see the attachment.

Reviewer 3 Report
This paper is of interest and although relevant to Finish population, the issues are likely to be the same in other countries with similar immigrant populations.
The study is conducted to a high standard with routinely collected data and the limitations are clearly discussed.
I understand the issue of levels of uptake of screening in different ethnic groups but does this mean that there have been concerns about the incidence of cervical cancer by ethinicity?
I have no major concerns and the analysis has taken into account potential confounders in this observational work.
Due to the data used, it is not possible to know the underlying issues and these can only be considered. It would be useful to consider further qualitative research to understand these in different ethnic groups and to consider potential strategies to promote participation. do the same groups fail to engage with other health care provision?
Author Response
Please, see the attachment.

Reviewer 4 Report
Abstract
Please explain how did you measure socioeconomic status.
Introduction
Please dedicate some words about how opportunistic screening influence the meaning of your results: do you have any idea of the difference between test coverage and participation in the program? Are there some differences in the uptake of opportunistic screening between immigrants and Finns? I think these aspects should be specified since the beginning.
The objective is ambiguous about the inclusion or not of opportunistic screening in computing the test coverage: “This study examined cervical cancer screening participation in the Finnish mass screening program, and factors associated with screening during 2008-2012 in women of Russian, Somali, and Kurdish origin compared with the general Finnish population in Finland.” In this sentence the authors introduce a primary endpoint unequivocally as “participation in mass screening”; then they introduce “factors associated with screening” in this second sentence screening can be interpreted as screening test coverage. If you refer to any way of test uptake please be explicit, if you refer only to screening within the mass screening program (as it seems clear from the methods), please refer to it as participation consistently.
Methods
Please define precisely participation: what is the time lapse too consider a woman as respondent to an invitation?
I suggest not adopting a significance threshold in this study: many comparisons are presented more with a descriptive intent than for testing an hypothesis, the sample size has not been formally dimensioned to make all these tests of hypothesis, but only for one main objective; also for the only question that is formulated as a test of hypothesis (i.e. comparing participation between immigrants and Finns) I do not think you can say that a p=0.051 brings you too accept the null hypothesis and a p=0.049 let you refuse the null hypothesis. P values and intervals should be interpreted in continuous.
Results
I suggest to avoid emphasis on statistical significance.
Discussion
It would be interesting to discuss the following findings in the light of some background of the history of immigration and of the health system organization:
- Time since immigration has different effect in different groups
- Contacts with hospitals increases participation despite the presence of an opportunistic offer: health system contacts do not bring to opportunistic screening even when finding women that are underscreened. Who is the target of opportunistic screening?
- Effefct of education is stronger in Finns than in other groups. This phenomenon has been observed in other countries and also for other health problems.
- Moving and having to stay abroad have different effects on different groups…
I suggest to discuss the low participation in Somali women in the light of the underlying risk of this group, both from data on cervical cancer incidence in Finland and other Nordic countries and from data on cervical cancer incidence and mortality in Somalia (data from the HPV information centre).
I think that of knowing if women are covered by opportunistic screening is a limitation of the study and should be highlighted and implications explained.
Check reference numbers (11 is wrong for sure)
Author Response
Please, see the attachment.
